# HUMAN-PRODUCIBLE ADVERSARIAL EXAMPLES

## ABSTRACT

Visual adversarial examples have so far been restricted to pixel-level image manipulations in the digital world, or have required sophisticated equipment such as 2D or 3D printers to be produced in the physical real world. We present the first ever method of generating human-producible adversarial examples for the real world that requires nothing more complicated than a marker pen. We call them *adversarial tags*. First, building on top of differential rendering, we demonstrate that it is possible to build potent adversarial examples with just lines. We find that by drawing just 4 lines we can disrupt a YOLO-based model in 54.8% of cases; increasing this to 9 lines disrupts 81.8% of the cases tested. Next, we devise an improved method for line placement to be invariant to human drawing error. We evaluate our system thoroughly in both digital and analogue worlds and demonstrate that our tags can be applied by untrained humans. We demonstrate the effectiveness of our method for producing real-world adversarial examples by conducting a user study where participants were asked to draw over printed images using digital equivalents as guides. We further evaluate the effectiveness of both targeted and untargeted attacks, and discuss various trade-offs and method limitations, as well as the practical and ethical implications of our work. The source code will be released publicly.

## 1 INTRODUCTION

Machine Learning (ML) has made significant progress over the past decade in fields such as medicine (Sidey-Gibbons & Sidey-Gibbons, 2019), autonomous driving (Jain et al., 2021), and biology (Zitnik et al., 2019). Yet it is now known to be fragile and unreliable in real-world use-cases (Goodfellow et al., 2015a; Biggio et al., 2013). A decade ago, ML models were discovered to be vulnerable to adversarial perturbations – small imperceptible changes that can mislead a given model and give control to an attacker (Carlini & Wagner, 2017). Such perturbed data are called *adversarial examples*. Ten years after their discovery, they are still a real threat to ML.

Up until recently, adversarial examples were mostly restricted to the digital domain, and bringing them to the real world presented significant challenges (Sun et al., 2018; Athalye et al., 2018). Although some work has demonstrated real-world adversarial examples, all of these approaches required specialized tools, *e.g.* 2D/3D printers or even specialized clothing (Ahmed et al., 2023), or applying to specific changes to objects (Eykholt et al., 2018). This need for special changes arises from the nature of traditional adversarial perturbations: imperceptible changes are too fine for humans to apply directly, while more visible examples were previously complex for humans to reproduce reliably without special resources (Brown et al., 2018). This significantly restricted their applicability in settings where no special resources are available.

In this paper, we revisit adversarial examples to make them more easily producible by humans. We devise a drawing method that makes perturbations visible and easily applied by humans. Our method is simple: it relies on drawing straight lines onto existing images or surfaces, a common skill that requires no training or advanced equipment (Cole et al., 2008). We call the collection of lines produced to target a given input image an *adversarial tag*, inspired by the art of graffiti.

In particular, we demonstrate that line-based adversarial tags are easy to produce and that they are as potent as their imperceptible counterparts. Next, inspired by research into human drawing, we devise a method to take human error into account when generating adversarial tags. Some examples are presented in Figure 1. We show that to reliably control a recent YOLO model, in over 80% of

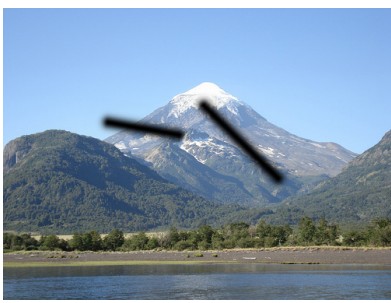

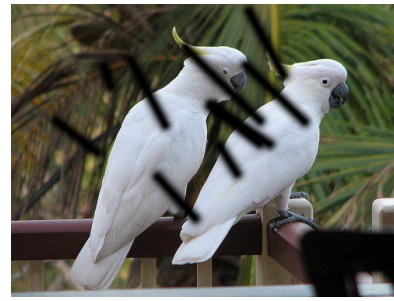

(a) **Before attack**: volcano (63.1%)
**After attack**: sundial (43.7%)
**Parameters**: 2 lines; 3,000 steps

(b) **Before attack**: cockatoo (100.0%)
**After attack**: albatross (53.1%)
**Parameters**: 10 lines; 10,000 steps

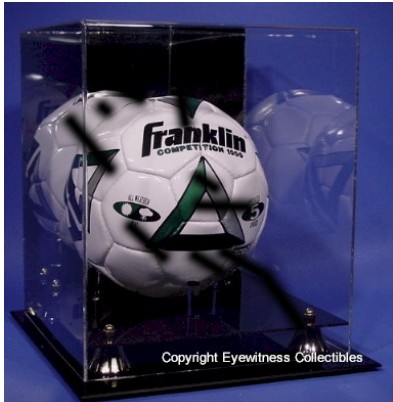

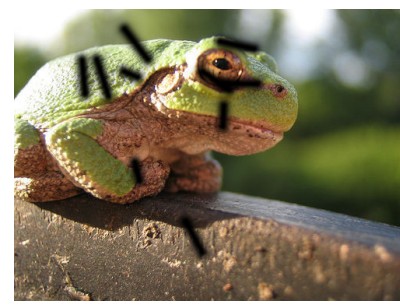

(c) **Before attack**: soccer ball (48.1%)
**After attack**: football helmet (42.4%)
**Parameters**: 10 lines; 10,000 steps

(d) **Before attack**: tree frog (52.4%)
**After attack**: chameleon (53.2%)
**Parameters**: 10 lines; 5,000 steps

Figure 1: Examples of generated adversarial examples. The predicted classes before and after the attack and primary algorithm parameters are specified for each example.

cases, an attacker needs to draw just 9 lines. We evaluate our method using extensive human trials to verify that it transfers into the physical world for both printed–scanned and photographed objects.

In summary, we make the following contributions:

- We present the first method of generating adversarial examples that can be produced by a human in the real world with nothing but a marker.
- We evaluate the effectiveness of our attack in both the digital and physical worlds and under targeted and non-targeted settings.
- We run a user study and discover that just as digital simulations suggest, humans are capable of reproducing adversarial tags with the necessary precision to make them effective.

## 2 BACKGROUND

### 2.1 ADVERSARIAL EXAMPLES

Adversarial examples can be defined as maliciously crafted inputs to ML models that mislead them, resulting in a non-obvious misclassification of the input. They were discovered and documented in 2013 by two separate teams led by Szegedy et al. (2013) and Biggio et al. (2013). We concern ourselves with a white-box environment, where an adversary has direct access to the model. Such examples are found using various gradient-based methods that aim to maximize the loss function under constraints (Goodfellow et al., 2015b; Carlini & Wagner, 2017).

## 2.2 PHYSICAL ADVERSARIAL EXAMPLES

Most adversarial examples rely on imperceptible changes to the individual pixel values of an image, with only some research into more noticeable examples, such as in the context of producing real-world adversarial objects. To the best of our knowledge, practically all the prior work required access to sophisticated equipment to apply their attacks such as data projectors or printers.

Some works produced adversarial examples projected onto a different representation. Sharif et al. (2019) crafted eyeglass frames that fooled facial recognition software. Wu et al. (2020); Xu et al. (2020) printed adversarial designs on t-shirts to let the wearers evade object-detection software. Komkov & Petiushko (2021); Zhou et al. (2018) fabricated headwear, such as baseball caps, to achieve similar results. Stuck-on printed patches and rectangles were investigated by Thys et al. (2019); Eykholt et al. (2018) and were shown to be effective. Ahmed et al. (2023) manufactured tubes that, when spoken into, cause ML-based voice authentication to break.

Given the rise of ML in daily life in fields that infringe on privacy, such as facial recognition (Wang et al., 2022) and other forms of surveillance (Liu et al., 2020), we wondered whether it would be possible to simplify the production of adversarial examples so that attacks did not require the creation of new objects, but just the modification of existing ones by graffiti. By democratizing the production of real-world adversarial examples, we hope to highlight the fragility of AI systems and call for more careful threat modeling of machine learning in the future.

Related to our work, Eykholt et al. (2018) used black and white patches on top of the objects, which can be considered thick lines. In contrast, we explicitly design our attack to be easily applicable by humans, show that it is realizable with lines placed outside of objects, and evaluate its efficacy with human experiments. In practice, robustification from Eykholt et al. (2018) can be used in conjunction with the attacks described in our work to launch more potent attacks against ML systems.

## 2.3 PRECISION OF HUMAN DRAWINGS

Developing adversarial tags that can be easily reproduced by humans requires understanding how people draw and the kind of errors they make. Our focus is on developing a method that works without any professional training or specialized tools.

Cole et al. (2008) provide a characterization of which pixels drawn by a line drawing algorithm are found in human line drawings. This work also notes that humans (concretely, artists) are consistent when drawing a subject – approximately 75% of human drawing pixels are within 1mm of a drawn pixel in all other drawings (from the study, for a specific drawing prompt). Carson et al. (2013) provide a characterization of human drawing error, including results from both novice and professional artists. While their work focuses primarily on characterizing the error in drawing polygons, the errors they noted can be extrapolated to parabolic lines. There are four main error types: orientation, proportionality, scaling, and position. Tchalenko (2009) quantify drawing accuracy for lines. Line shape was found to be the largest contributor to overall error, followed by the overall size. Proportions were often off by a factor of 20-30%, but this was a smaller error. Curiously, Grossi et al. (1998) find that some people cannot draw horizontal lines, while their ability to draw vertical ones is unimpaired. This suggests that mental representations of horizontal and vertical spatial relations in an egocentric coordinate system are functionally dissociated.

In this paper, we rely on humans' inherent ability to draw straight lines to produce effective adversarial tags. Since the literature reports that humans still produce minor line placement errors, we model this in our adversarial generation loop. A visual example of the allowable error margins in human drawing that we account for is presented in Figure 2. We do not explicitly limit the use of horizontal lines, since we found in user studies that all participants were nevertheless still capable of producing working adversarial examples.

## 3 METHODOLOGY

### 3.1 LINE PLACEMENT

In contrast to the classic adversarial example generation literature, where the perturbations are derived directly from gradient calculations, our restricted setting requires careful consideration of ini-

tial line positioning. We use a *generate-and-prune* approach inspired by Cun et al. (1990). The algorithm also bears a resemblance to genetic algorithms but is fundamentally a hybrid approach between gradient methods and more computationally-intensive gradient-free ones (Chahar et al., 2021). We build up a collection of lines, up to a predefined maximum collection size of $N$, by iteratively performing the following every $m$ steps (with $m = 100$ unless stated otherwise):

1. Generate $f$ random lines, where $f$ is a given expansion factor. Unless stated otherwise, we take $f = 10$.
2. Prune the joined set of generated lines and the existing collection of lines. The top $k$ candidates would be retained as the new collection of adversarial lines.

Pruning is then done as follows:

1. The operation starts with a collection of lines of size $c$. The *generate* step outlined above brings the collection up to size $c + f$.
2. For each line, calculate the mean of the absolute gradient values of its four line-defining parameters (*start/end x/y coordinates*). The gradient values are calculated via a backward pass w.r.t to the line parameters and using a loss similar to that described in the next subsection. Take the top $k = c + 1$ candidates, based on the above metric, to be the new collection of lines to be applied to the image.

## 3.2 ROBUST LOSS

Since the focus of this work is to enable *human-producible* adversarial examples, we need to allow for the drawing errors that humans make – in orientation, proportionality, scaling, and position – as identified by Carson et al. (2013). We do this by allowing for a jitter in both the start and end coordinates of the lines, controlled by a jitter factor $j$ (usually $j = 0.05$). We also introduce an erasure factor $e$ (usually $e = 0.25$): when drawing the lines, a percentage of drawn pixels are zeroed out. The magnitude of these errors is visualized in Figure 2. This assumes imperfections when a human produces the adversarial example, in both the scanning technology used to digitize the sample and the drawing implements used to produce it.

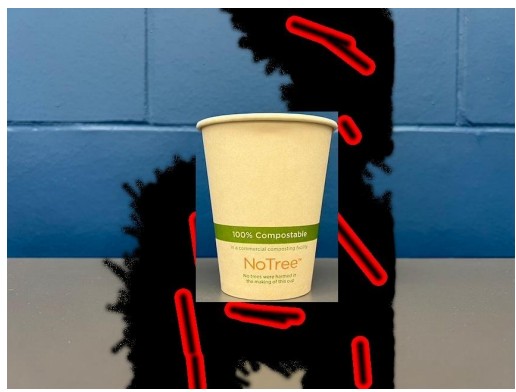

Figure 2: Example of lines jittered and erased to account for human drawing error. In red we show the final perturbation lines, while in black we demonstrate the range of error we account for.

These two stochastic factors – jitter and erasure – are accounted for by generating a fixed number of auxiliary lines $n$ (usually $n = 4$) at each step when calculating the loss. When generating *non-robust* adversarial examples, $n$ is set to 1 and the jitter $j$ and erasure $e$ factors are 0.

We incorporate these concepts in what we call a *robust loss*. The main loss calculation is detailed in Algorithm 1. This takes into account the human errors discussed above. We are able to directly optimize the line parameters by making use of the work by Mihai & Hare (2021) to achieve auto-differentiable rasterization of lines. This greatly simplifies and speeds up the generation process.

This robust loss can then be optimized to produce an adversarial example for a given image. This can be done in both a targeted and untargeted fashion. For untargeted attacks, the target class $t$ is taken to be the originally predicted class on the input image $\mathbf{I}$, and the loss sign is flipped.

## 3.3 METHOD

As detailed in Section 3.1, we iteratively build up a collection of adversarial lines to form the adversarial tag. These are conditioned differently depending on whether we optimize for a targeted or

---

**Algorithm 1** Calculating robust loss for collection of lines

---

**Input:** line parameters l, jitter $j$ and erasure $e$ factors, image dimensions $s$, number of auxiliary lines per line $n$, input image $\mathbf{I}$, target class $t$
Initialize total loss $\mathcal{L}$
**for** $i = 1$ **to** $n$ **do**
    $\mathbf{l}' \leftarrow \text{clamp}(\mathbf{l} + s \times \mathcal{U}(-j/2, j/2), 0, s)$
    $\mathbf{L} \leftarrow \text{random\_erase}(\text{render\_lines}(\mathbf{l}'))$
    $\mathcal{L} \leftarrow \mathcal{L} + \text{negative\_log\_likelihood\_loss}(\mathbf{I} + \mathbf{L}, t)$
**end for**
**Output:** total loss $\mathcal{L}$

---

untargeted attack. We also can control the level of robustness in the loss that we optimize for, as previously described. We evaluate both robust and non-robust loss – while the former is shown to produce better results in the user study, the latter is significantly faster to generate.

We keep track of the best loss – defined as the largest for an untargeted attack for the original class, but the smallest for a target class for a targeted attack. Since the generation process is stochastic, we need to allow for backtracking: if the best parameters are not updated after a set number of steps (usually 1000), the parameters are reset to these ones, and the optimization process continues. If no further progress is made after four such resets, the optimization terminates. Otherwise, it terminates after a fixed number of steps (usually 10,000). The number of lines specified at the start of the optimization process is a strict maximum, and the best adversarial line set may use fewer.

### 3.4 USER STUDY

For a quantitative evaluation of the real-world effectiveness of the adversarial tags we generate, we conducted a user study with four participants. We received an approval from the University Ethics Board and closely followed the guidelines for human studies. Each participant was presented with four sets of the same 20 unique image collages to modify using a black marker. Each image collage consisted of four individual images: the original unmodified image to be used as a baseline, a black-on-white rendering of the lines to draw, the generated adversarial sample (that is, the lines superimposed on the original image), and the original unmodified image to be drawn on by the participant. Participants were selected without any artistic ability bias and were asked to self-evaluate their drawing ability before the experiment began. This included questions relating to any formal training received and frequency of practice. After each image, the participant noted how long they spent drawing the lines and a self-evaluation of the difficulty of tracing the lines by eye.

The 4 sets consist of different approaches to generating the adversarial lines – untargeted non-robust, untargeted robust, targeted non-robust, and targeted robust. The target classes were chosen randomly when initially generating the adversarial examples. This approach allowed us to evaluate both the variance between users for the same image, and the variance between the different approaches when it comes to human performance. The pages were scanned and then automatically cropped to size. The original unmodified baseline image, and the modified image with hand-drawn lines, were extracted from the scanned pages. These processed images were then run through YOLOv8 (Jocher et al., 2023) to obtain confidence values.

### 3.5 PRACTICAL USE-CASES

With adversarial tags, it is obvious that the images have been tampered with as the lines are prominent to the human eye. Humans can recognize incomplete or adversarially augmented shapes and figures, including text. ML models are not yet as capable, and the difference has been exploited for years by the designers of CAPTCHAs. The gap is becoming an issue for ML systems: while a stop-sign street sign with a few graffiti marks will not be given a second look by a human driver, it can be easily misclassified by a driver-assistance system if those marks were made adversarially (Rawlinson, 2007; Ayzenberg, 2019; Biederman et al., 1982).

To test this real-world scenario, we conducted an experiment whereby we took photographs of a common household object – a cup depicted in Figure 6a – and produced an adversarial tag for it.

We constrained the search area to a rectangular bounding box to limit the lines to a specific area of the image to avoid the cup itself. We then recreated the lines using black tape and re-took the photographs. Results are presented in the following section.

## 4 EVALUATION

We evaluate on the `YOLOv8n-cls` classification model (Jocher et al., 2023) and the `ImageNet` dataset (Deng et al., 2009). `ImageNet` was chosen as it is one of the most diverse and commonly used image classification datasets. Many of the images in it are photos of real objects, making them suitable targets for graffiti. This also means that we humans are pre-trained models to carry out the evaluation and match the common realistic setup where YOLO, a widely deployed object detection and classification model, is used out of the box. We ran experiments on a locally-hosted GPU server with `4×NVIDIA RTX 2080Ti` cards, each with approximately 11 GB of GPU memory, using the PyTorch (Paszke et al., 2019) ML framework.

The iterative nature of the algorithm, and the computationally intensive nature of back-propagating directly over the line parameters, mean that our method of generating adversarial examples, especially with robust loss, is time-consuming and compute-expensive. With this in mind, and due to our limited computing resources and with ecological considerations in mind, we evaluate a random sample of 500 images drawn from `ImageNet`'s validation set. The main metric we use for evaluation is the notion of whether the top-1 predicted class changed – *i.e.* whether the class was *flipped*. That is, we measure if the class assigned the highest probability for a given image changes after the application of the adversarial tags. Hence, we report the ratio of images *flipped* in our test dataset.

### 4.1 LINE PARAMETERS

While the method described can be used to optimize over Bézier curve parameters, we exclusively use 'straight' lines due to their simplicity and ease of production. We stick to black-colored lines as black ink overlays well over all colors. Once the lines are rasterized, the rendered pixels are simply subtracted from the image with range clamping. Line thickness is controlled by a parameter $\sigma$, which was set arbitrarily to 60 to visually match the thickness of a standard whiteboard marker on A4 paper when the images were printed out for our user study.

The optimal characteristics of the lines required to produce satisfying results were investigated. The main trade-off was found to be between generating a large number (20-40) of shorter lines, and fewer ($\leq$12) longer lines. The former approach gave marginally better results, but we considered it impractical for human users to draw many lines quickly and accurately without tools such as rulers or stencils. This impracticality was confirmed via the user study.

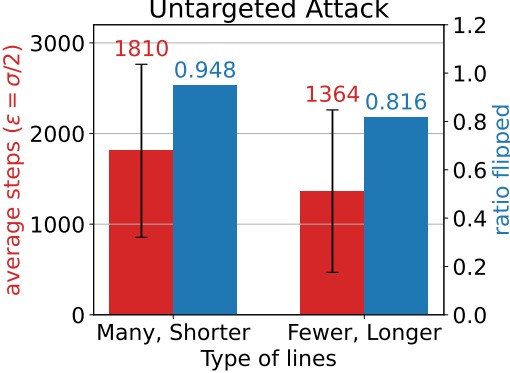

Figure 3: Comparing success rates of *many shorter* lines (defined to be lines of length 20-50px, numbering between 25-35) vs *fewer longer* lines (defined to be lines of length 80-120px, numbering between 8-12).

Detailed experiments regarding this trade-off are presented in Figure 3. We can see similar performance for both groups, but with the *fewer longer* group taking nearly 25% fewer steps with a factor of $3 - 4$ fewer lines which results in significant compute saving and easier human reproduction.

### 4.2 IMAGENET

We conduct experiments to gauge the effectiveness of our proposed method for flipping the predicted image class in both an untargeted and targeted manner.

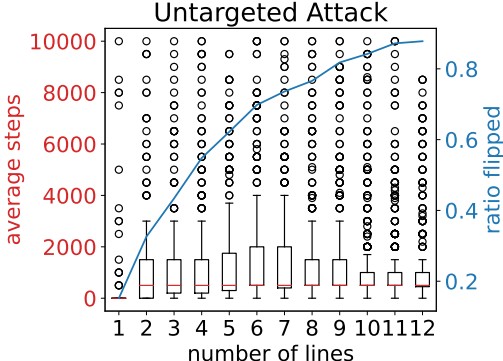 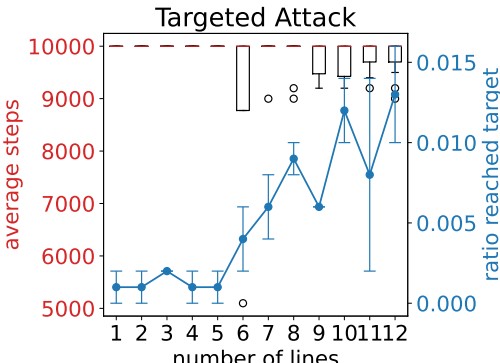

Figure 4: This figure concerns an *untargeted* attack and shows the number of adversarial lines against two metrics. The blue line shows the percentage of tested images that had their top-1 prediction changed (*i.e. flipped*) within 10000 steps for a given number of lines. The red line shows the average number of steps it took to achieve this flip, together with the standard deviation.

Figure 5: This figure concerns a *targeted* attack and shows the number of adversarial lines against two metrics. The blue line shows the percentage of tested images that had their top prediction changed to the target class within 10,000 steps for a given number of lines. The red line shows the average number of steps it took. The results are obtained from two runs over the same dataset with randomly selected targets.

The untargeted results are presented in Figure 4. We can see a trend whereby increasing the number of adversarial lines used increases the ratio of images with a flipped class. These range from 15.2% for 1 line, to 54.8% for 4 lines, 81.8% for 9 lines, and 87.8% for 12 lines. As can be seen in the figure, the number of steps required to achieve the class flip remains more or less constant throughout, with a relatively large standard deviation, owing to the diversity of images in the test set. We also report attack norms in Appendix B, and present a transferability case-study in Appendix C, where we show that attacks also posses some transferability across models in the Yolo family.

Figure 5 presents results for targeted attacks on random targets. Rather than measuring the ratio of flipped classes, we present the ratio of samples that reached their intended target class. While increasing the number of lines helped with this goal, we can see that performing targeted attacks is challenging. We hypothesize that the reason is two-fold. First, targeted attacks present a harder optimization task due to increased constraints, and the allowed search space of black straight lines is not flexible enough to accommodate this. Secondly, we find that for targetting to work, application of the lines has to be very precise and minor changes in the input cause the output to change. It is worth noting that while a sample might not reach the intended target, we anecdotally find that it often reaches a semantically similar one. For example, if the target class was `tarantula (76)`, the adversarial image might end up classified as `barn spider (73)` after optimization.

### 4.3  USER STUDY

The user study generated a total of 320 image collages, each consisting of four individual images, as described in the methodology section. The baseline images were reclassified after being scanned and it was found that 65.9% of the images retained their original class. We filtered out the image collages that did not retain the original class and presented data only based on the remaining samples. We report time taken to reproduce the tags in Appendix D.

We present two separate analyses of the data. The first analysis has the data grouped by attack type (un/targeted) and loss type (non/robust). The second one groups the data by the number of lines, namely looking at low line counts [3, 7] inclusive, versus high line counts [8, 12] inclusive. We can clearly see the effect of robust loss on improving human reproducibility by comparing the percentage class change of the samples scan-to-scan when the scanned baseline image retained its original class, and the initial digital images had a class change – *i.e.* were successfully flipped by the adversarial lines. For untargeted attacks, this percentage stood at 46.2% for non-robust lines, and 77.8% for robust lines – almost a 70% increase in reliability of human reproduction.

|  | Number of lines | Retained new class after scanning | Class change scan-to-scan |
|---|---|---|---|
| Non-Robust | [3, 7] | 0.0% | 62.5% |
|  | [8, 12] | 13.0% | 34.8% |
| **Robust** | **[3, 7]** | **50.0%** | **100.0%** |
|  | [8, 12] | 15.8% | 68.4% |

Table 1: User study results for an untargeted attack, grouped by number of adversarial lines. All results are presented with the precondition that the baseline image retained its original class and the digital pair had a class change.

We can also compare the percentage of samples that retained their new class – as defined to be the class assigned to the digital image after the application of adversarial lines – after scanning, given the baseline image retained the original class and the digital image pair had a class change. For untargeted attacks, this figure was measured to be 7.7% for non-robust lines and 25.9% for robust ones. This is over a factor of 3 increase. It is worth noting that for an untargeted attack, new class retention is not as important as whether the class flipped scan-to-scan. Hence we can conclude that the robust loss significantly improves human reproducibility. Targeted attacks, as previously discussed, were not found to work particularly well both digitally and in the physical world.

We then turn to consider adversarial tag performance grouped by the number of lines. The results are presented in Table 1. First, we note that robust examples outperformed non-robust in terms of human reproducibility. However, the remaining results are surprising and contrary to the ones obtained in the digital realm presented in Figure 4 – we observe that the class change scan-to-scan gets *worse* with more lines, as does new class retention for robust lines. We hypothesize this happens due to humans finding it difficult to accurately reproduce larger numbers of lines. This confirms our assumptions outlined in the *line parameters* section regarding optimizing for fewer longer lines to improve human reproducibility.

## 4.4 CASE STUDY IN THE REALITY: PAPER CUP

Finally, we conduct a user study of replicating human-producible adversarial examples in the real world. The user study includes a controlled environment around a paper cup (shown in Figure 6a) and 7 participants. We present the printed adversarial example to each participant and ask them to replicate the lines by applying tapes to the corresponding locations. All participants are college students aged between 20 and 28 without training experience related to artwork or this task. We did not use a marker pen because the environment needs to be recovered after each participant's application and the potential for cross-contamination between the users. Notably, tapes have a similar appearance to marker drawings and even better accessibility for humans. Importantly, they also allow for direct comparison to related work (Eykholt et al., 2018).

Figure 6 shows the original environment, the adversarial examples presented to the participant, and the participant's replication. We can see that the replicated adversarial examples successfully disrupt the model's predictions in all of the replications. We present more user replications of the non-robust and robust adversarial examples in Figures 7 and 8 in the Appendix, where the replications remain adversarial even if participants have applied tapes with noticeable errors. These results confirmed that the adversarial tags are easily reproducible by humans and are robust to non-precise replication.

## 5 DISCUSSION

**Ethical implications** Adversarial tags, as defined here, have serious ethical and societal implications. The capability to disrupt the functionality of advanced object-detection models using minimalistic and easily accessible tools, such as a marking pen, underscores the inherent vulnerability of real-world systems to manipulation. This raises concerns regarding the potential misuse of such techniques, including but not limited to evading surveillance, deceiving autonomous vehicles, or compromising security systems. Given the ease of use, human-producible tags need to be explicitly

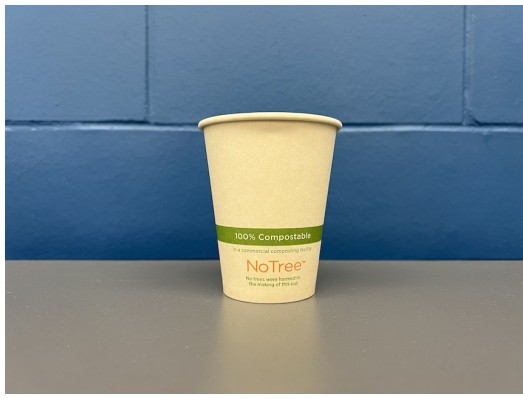

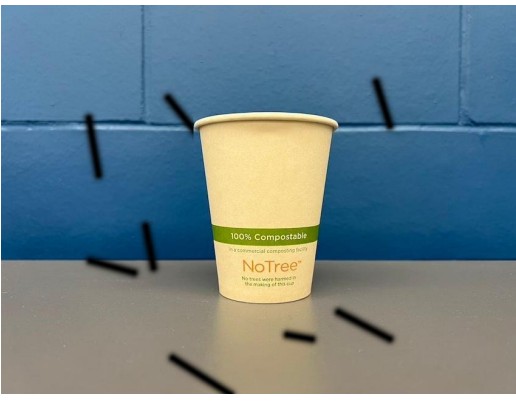

(a) **Original: beer glass 0.87**, beaker 0.03, lotion 0.02, sunscreen 0.01, measuring_cup 0.01.

(b) **Non-robust adversarial example: nail 0.66**, rule 0.09, measuring_cup 0.05, paintbrush 0.04.

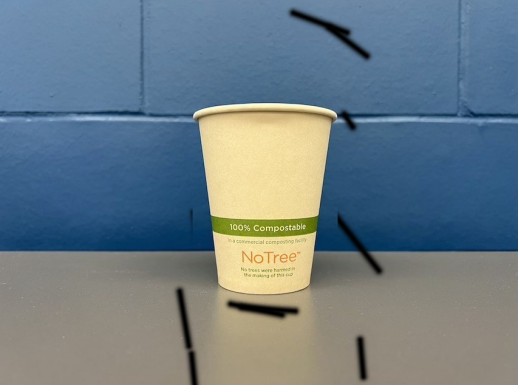

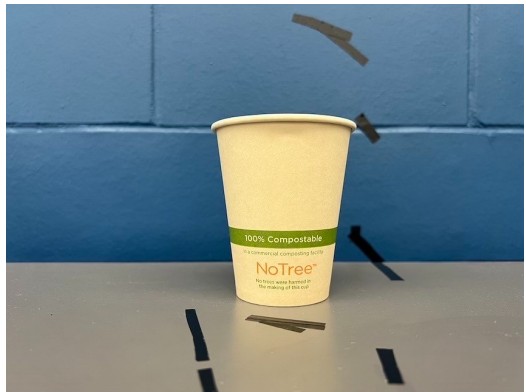

(c) **Robust adversarial example: paintbrush 0.38**, nail 0.26, syringe 0.05, screwdriver 0.04, beaker 0.03.

(d) **User replication: bucket 0.50**, ashcan 0.16, paintbrush 0.12, tennis_ball 0.02, carton 0.02.

Figure 6: The user study of replicating adversarial examples in the real world.

modeled for and taken into consideration. Otherwise, we may end up in situations where a Tesla may well change its intended route due to its sensors picking up an adversarial graffiti tag on a wall.

**Limitations of targeted attacks** In this work, we find that targeting specific classes is often challenging with adversarial tags. We associate this with a limited adversarial search space, which in turn is responsible for making the tags producible by humans. We hypothesise that by giving more degrees of freedom to change the lines one can control the output of the target class more efficiently, at a cost of human reproducibility *e.g.* by giving control over the color of the lines or inclusion of other shapes, colors, variable thickness.

## 6 CONCLUSION

In this paper we demonstrated that people can mark objects with adversarial graffiti tags which cause these objects to be misclassified by machine vision systems. These tags can be applied reliably by people with little to no supervision and training, using just a few straight lines copied from a sketch. This technique can have applications from privacy to camouflage, and raises interesting questions about robustness standards for systems that incorporate machine vision models.

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

## A   CASE STUDY IN THE REALITY: PAPER CUP

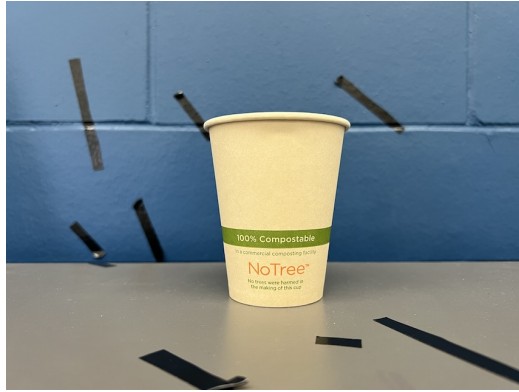

(a) **User 1**: paintbrush 0.50, nail 0.23, bucket 0.09, rule 0.04, beaker 0.03

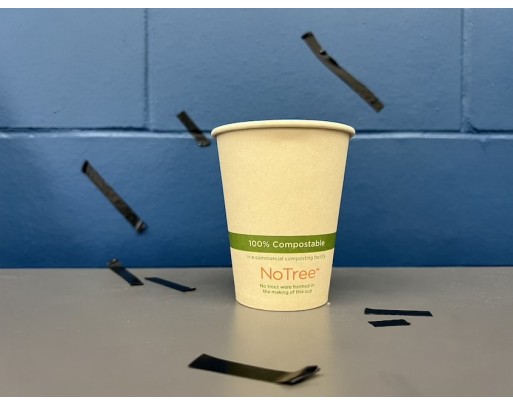

(b) **User 2**: nail 0.64, matchstick 0.16, bucket 0.06, paintbrush 0.06, screwdriver 0.02

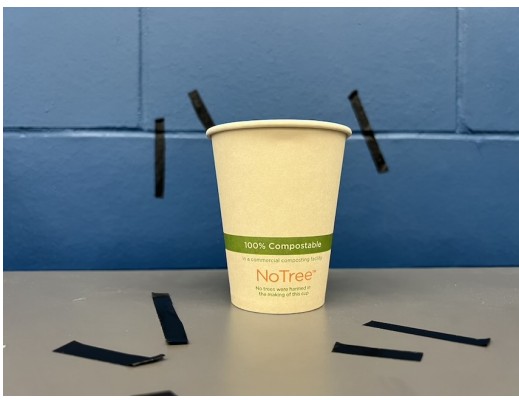

(c) **User 3**: bucket 0.62, paintbrush 0.10, rule 0.05, nail 0.04, beaker 0.04

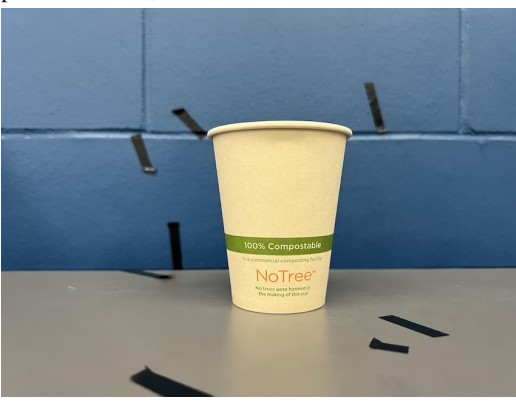

(d) **User 4**: bucket 0.59, ashcan 0.10, beer glass 0.03, paintbrush 0.03, measuring cup 0.02

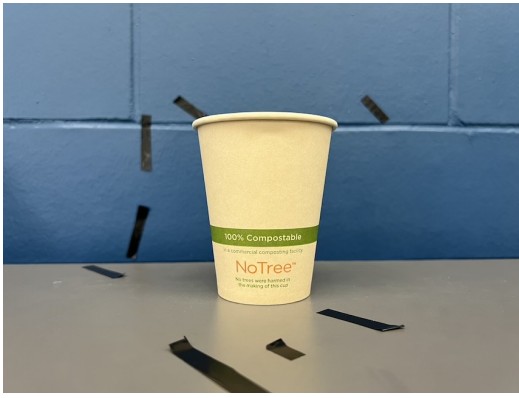

(e) **User 5**: bucket 0.47, paintbrush 0.18, ashcan 0.07, carton 0.06, nail 0.03

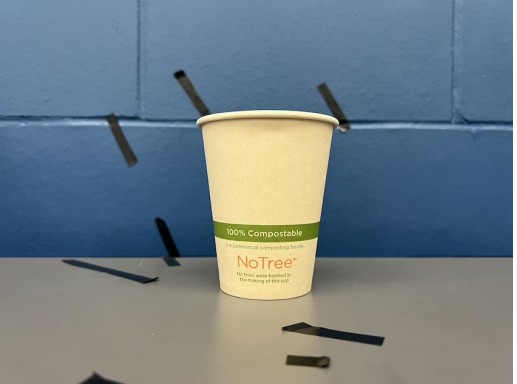

(f) **User 6**: bucket 0.48, nail 0.09, paintbrush 0.08, carton 0.04, ashcan 0.03

Figure 7: User replications of **non-robust** adversarial examples.

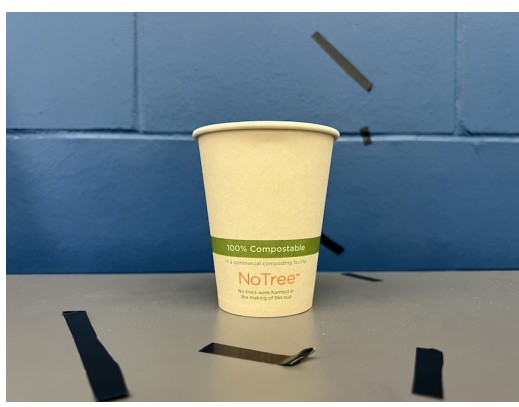

(a) **User 1**: bucket 0.44, ashcan 0.33, paintbrush 0.02, beer glass 0.01, carton 0.01

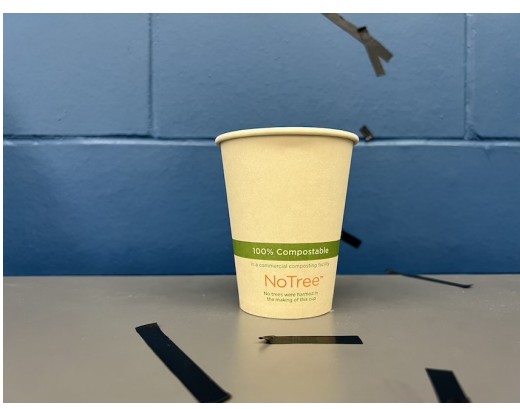

(b) **User 2**: bucket 0.63, paintbrush 0.12, ashcan 0.06, nail 0.01, carton 0.01

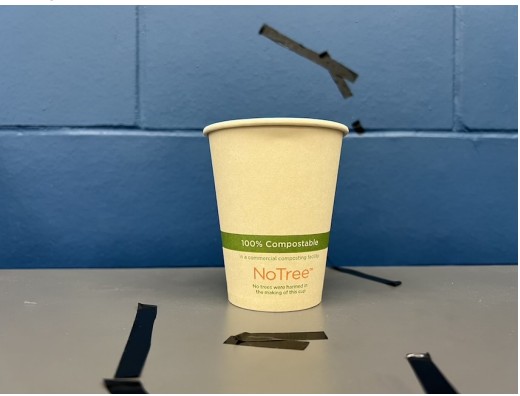

(c) **User 3**: ashcan 0.24, nail 0.22, bucket 0.17, paintbrush 0.07, hammer 0.03

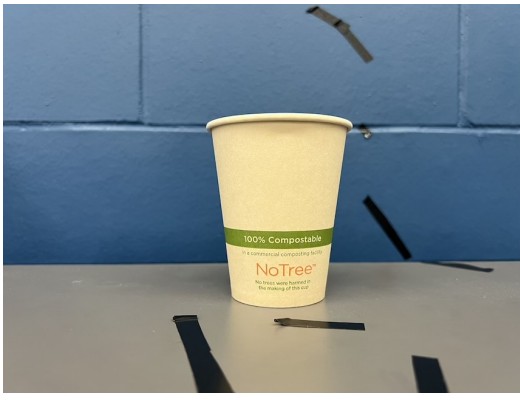

(d) **User 4**: bucket 0.43, paintbrush 0.16, nail 0.11, ashcan 0.07, hammer 0.02

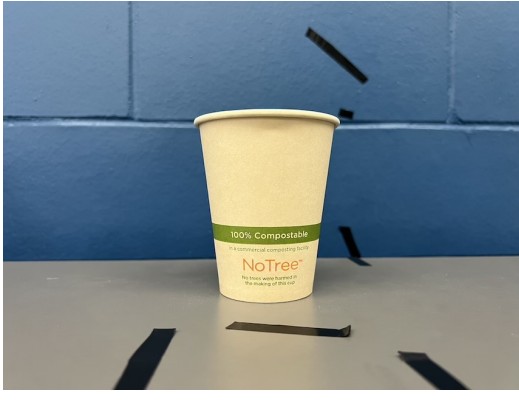

(e) **User 5**: nail 0.48, paintbrush 0.23, screwdriver 0.08, bucket 0.07, ashcan 0.02

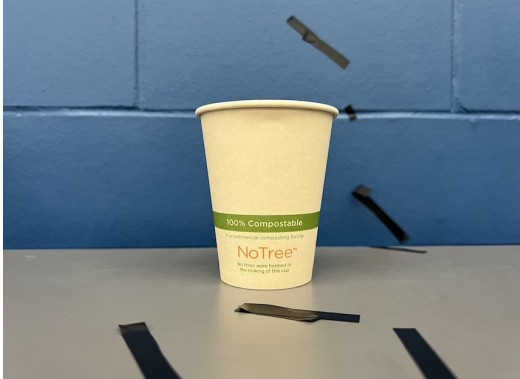

(f) **User 6**: nail 0.39, paintbrush 0.33, screwdriver 0.13, hammer 0.03, bucket 0.02

Figure 8: User replications of **robust** adversarial examples.

## B  $L_2$ AND $L_{INF}$ NORMS OF THE ATTACK

In Tables 3 and 5 we present the L2 and $L_{inf}$ norms of our adversarial tags, split between the targeted and untargeted attacks.

| Line Count | $L_2$ | $L_{inf}$ | Line Count | $L_2$ | $L_{inf}$ |
|---|---|---|---|---|---|
| 1 | 25.02 (9.56) | 0.84 (0.18) | 7 | 60.54 (20.38) | 0.95 (0.10) |
| 2 | 35.16 (11.86) | 0.90 (0.13) | 8 | 62.28 (20.83) | 0.95 (0.10) |
| 3 | 42.02 (13.13) | 0.93 (0.11) | 9 | 64.50 (22.77) | 0.95 (0.10) |
| 4 | 47.33 (15.46) | 0.93 (0.12) | 10 | 66.32 (25.02) | 0.95 (0.11) |
| 5 | 52.66 (17.21) | 0.94 (0.11) | 11 | 68.10 (25.58) | 0.95 (0.10) |
| 6 | 57.11 (18.61) | 0.95 (0.10) | 12 | 69.23 (27.08) | 0.95 (0.10) |

Table 2: Mean $L_2$ and $L_{inf}$ norms of *untargeted* adversarial tags. Standard deviation shown in parentheses.

| Line Count | $L_2$ | $L_{inf}$ | Line Count | $L_2$ | $L_{inf}$ |
|---|---|---|---|---|---|
| 1 | 24.64 (9.93) | 0.84 (0.18) | 7 | 66.76 (16.86) | 0.96 (0.08) |
| 2 | 35.43 (11.45) | 0.90 (0.14) | 8 | 71.34 (18.26) | 0.97 (0.07) |
| 3 | 44.35 (12.58) | 0.93 (0.11) | 9 | 75.20 (19.31) | 0.97 (0.07) |
| 4 | 50.50 (14.03) | 0.94 (0.09) | 10 | 78.31 (19.86) | 0.97 (0.06) |
| 5 | 56.39 (14.19) | 0.95 (0.08) | 11 | 82.85 (20.20) | 0.98 (0.06) |
| 6 | 62.43 (16.03) | 0.96 (0.08) | 12 | 85.30 (20.70) | 0.97 (0.07) |

Table 3: Mean $L_2$ and $L_{inf}$ norms of *targeted* adversarial tags. Standard deviation shown in parentheses.

## C  TRANSFERABILITY TO OTHER YOLO ARCHITECTURES

In the figures below we show targeted and untargeted attack transferability between different YOLO architectures. The model that was attacked was the smallest architecture `yolov8n`.

| № lines | *yolov8n* | yolov8s | yolov8m | yolov8l | yolov8x |
|---|---|---|---|---|---|
| 1 | 12.6% | 7.8% | 6.0% | 5.8% | 7.2% |
| 2 | 27.2% | 12.0% | 10.6% | 10.2% | 10.4% |
| 3 | 38.0% | 15.6% | 13.4% | 13.4% | 12.0% |
| 4 | 48.6% | 20.6% | 16.4% | 16.6% | 13.8% |
| 5 | 55.0% | 25.6% | 17.6% | 20.2% | 18.0% |
| 6 | 59.4% | 25.6% | 19.2% | 18.6% | 19.2% |
| 7 | 64.0% | 30.2% | 20.6% | 22.2% | 20.6% |
| 8 | 67.8% | 30.2% | 20.2% | 19.4% | 20.0% |
| 9 | 74.2% | 30.4% | 20.4% | 21.0% | 19.2% |
| 10 | 75.6% | 37.0% | 23.4% | 21.8% | 21.4% |
| 11 | 79.0% | 31.8% | 23.4% | 24.0% | 23.4% |
| 12 | 80.4% | 35.0% | 24.2% | 24.0% | 24.2% |

(a) Untargeted attack, showing percentage of attacked images where the original class was not the top 1 predicted class after applying the adversarial tag.

| № lines | *yolov8n* | yolov8s | yolov8m | yolov8l | yolov8x |
|---|---|---|---|---|---|
| 1 | 12.6% | 7.8% | 6.0% | 5.8% | 7.2% |
| 2 | 27.2% | 12.0% | 10.6% | 10.2% | 10.4% |
| 3 | 38.0% | 15.6% | 13.4% | 13.4% | 12.0% |
| 4 | 48.6% | 20.6% | 16.4% | 16.6% | 13.8% |
| 5 | 55.0% | 25.6% | 17.6% | 20.2% | 18.0% |
| 6 | 59.4% | 25.6% | 19.2% | 18.6% | 19.2% |
| 7 | 64.0% | 30.2% | 20.6% | 22.2% | 20.6% |
| 8 | 67.8% | 30.2% | 20.2% | 19.4% | 20.0% |
| 9 | 74.2% | 30.4% | 20.4% | 21.0% | 19.2% |
| 10 | 75.6% | 37.0% | 23.4% | 21.8% | 21.4% |
| 11 | 79.0% | 31.8% | 23.4% | 24.0% | 23.4% |
| 12 | 80.4% | 35.0% | 24.2% | 24.0% | 24.2% |

(b) Targeted attack, showing percentage of attacked images where the original class was not the top 1 predicted class after applying the adversarial tag.

| № lines | *yolov8n* | yolov8s | yolov8m | yolov8l | yolov8x |
|---|---|---|---|---|---|
| 1 | 2.8% | 2.4% | 0.8% | 0.8% | 1.2% |
| 2 | 5.2% | 3.0% | 3.0% | 1.4% | 2.6% |
| 3 | 8.0% | 4.4% | 3.8% | 4.2% | 3.2% |
| 4 | 11.2% | 5.8% | 4.4% | 5.0% | 3.8% |
| 5 | 14.4% | 9.4% | 6.0% | 5.6% | 5.2% |
| 6 | 16.2% | 10.4% | 8.2% | 6.6% | 7.4% |
| 7 | 20.8% | 13.4% | 9.2% | 7.2% | 9.2% |
| 8 | 19.6% | 13.2% | 8.8% | 7.6% | 8.2% |
| 9 | 23.0% | 14.6% | 8.8% | 8.2% | 8.0% |
| 10 | 26.2% | 14.4% | 7.4% | 7.6% | 10.0% |
| 11 | 26.2% | 14.2% | 11.2% | 9.0% | 10.0% |
| 12 | 27.0% | 15.6% | 10.2% | 8.8% | 11.0% |

(c) Untargeted attack, showing percentage of attacked images where the original class was not in the top 3 predicted class after applying the adversarial tag.

| № lines | *yolov8n* | yolov8s | yolov8m | yolov8l | yolov8x |
|---|---|---|---|---|---|
| 1 | 1.5% | 1.2% | 0.7% | 1.2% | 0.9% |
| 2 | 5.0% | 4.1% | 2.2% | 2.3% | 3.1% |
| 3 | 7.0% | 5.4% | 3.8% | 3.4% | 3.7% |
| 4 | 10.4% | 7.0% | 6.2% | 5.1% | 6.0% |
| 5 | 14.9% | 10.5% | 7.0% | 7.3% | 7.1% |
| 6 | 16.1% | 11.6% | 9.1% | 8.1% | 8.9% |
| 7 | 19.7% | 15.5% | 10.8% | 8.9% | 10.4% |
| 8 | 23.0% | 17.2% | 13.8% | 12.3% | 13.4% |
| 9 | 24.5% | 19.6% | 14.4% | 12.4% | 13.4% |
| 10 | 26.7% | 21.0% | 15.1% | 13.5% | 14.3% |
| 11 | 30.5% | 23.5% | 17.1% | 16.5% | 16.3% |
| 12 | 32.2% | 25.9% | 17.9% | 17.5% | 17.8% |

(d) Targeted attack, showing percentage of attacked images where the original class was not in the top 3 predicted class after applying the adversarial tag.

| № lines | *yolov8n* | yolov8s | yolov8m | yolov8l | yolov8x |
|---|---|---|---|---|---|
| 1 | 1.2% | 1.8% | 0.2% | 0.4% | 0.6% |
| 2 | 3.2% | 2.2% | 1.0% | 0.6% | 1.2% |
| 3 | 3.8% | 3.2% | 2.6% | 2.8% | 2.6% |
| 4 | 6.0% | 4.2% | 1.8% | 2.4% | 2.0% |
| 5 | 8.6% | 7.0% | 4.0% | 2.8% | 2.8% |
| 6 | 10.8% | 7.2% | 5.8% | 5.0% | 4.8% |
| 7 | 12.0% | 8.2% | 5.2% | 5.2% | 6.2% |
| 8 | 11.8% | 8.4% | 6.2% | 3.8% | 5.6% |
| 9 | 13.4% | 8.2% | 6.6% | 4.8% | 5.8% |
| 10 | 15.0% | 9.8% | 5.8% | 5.6% | 6.0% |
| 11 | 16.6% | 11.0% | 6.4% | 5.2% | 7.4% |
| 12 | 17.0% | 9.6% | 6.8% | 6.0% | 7.8% |

(e) Untargeted attack, showing percentage of attacked images where the original class was not in the top 5 predicted class after applying the adversarial tag.

| № lines | *yolov8n* | yolov8s | yolov8m | yolov8l | yolov8x |
|---|---|---|---|---|---|
| 1 | 0.7% | 0.4% | 0.1% | 0.5% | 0.5% |
| 2 | 2.9% | 2.2% | 1.6% | 1.7% | 1.8% |
| 3 | 4.2% | 3.8% | 2.7% | 2.4% | 1.8% |
| 4 | 6.8% | 4.9% | 3.9% | 3.3% | 3.1% |
| 5 | 8.7% | 7.6% | 5.1% | 4.1% | 4.6% |
| 6 | 10.0% | 8.6% | 5.9% | 5.4% | 5.6% |
| 7 | 13.9% | 10.2% | 7.6% | 7.1% | 6.7% |
| 8 | 16.4% | 13.0% | 10.6% | 8.6% | 9.4% |
| 9 | 17.2% | 14.7% | 11.1% | 9.1% | 10.1% |
| 10 | 19.4% | 14.8% | 11.1% | 9.5% | 10.4% |
| 11 | 23.7% | 16.6% | 12.9% | 13.0% | 12.5% |
| 12 | 24.4% | 20.8% | 13.0% | 13.2% | 13.7% |

(f) Targeted attack, showing percentage of attacked images where the original class was not in the top 5 predicted class after applying the adversarial tag.

Table 4: Transferability of adversarial tags between YOLO architectures. The model that was attacked was the smallest architecture `yolov8n`.

## D  User Study Timings

In the table below, we present the mean and standard deviation of the self-reported times that participants took to recreate the adversarial tags as described in Section 4.3.

| Line Count | Time [s] | Line Count | Time [s] |
|:---:|:---:|:---:|:---:|
| 3 | 6.13 (3.48) | 8 | 16.88 (8.44) |
| 4 | 8.13 (5.36) | 9 | 15.88 (7.40) |
| 5 | 10.13 (5.64) | 10 | 14.63 (7.93) |
| 6 | 10.75 (6.14) | 11 | 14.63 (10.10) |
| 7 | 10.13 (4.55) | 12 | 18.63 (11.01) |

Table 5: Self-reported time to reproduce adversarial tags, based on the total numbers of line present in the tag. Measured in seconds, with standard deviation shown in parentheses.

