# OpenReview forum: "Human-Producible Adversarial Examples"
_ICLR.cc/2024/Conference — Submitted to ICLR 2024_

### Official Review · Reviewer_Xsun · 2023-10-20

**Soundness:** 3 good
**Presentation:** 3 good
**Contribution:** 2 fair
**Rating:** 3
**Confidence:** 5

**Summary:**

This paper argues that existing manipulations for creating physical adversarial examples are hard to be reproduced by humans (without access to an electronic device). Therefore, they propose to generate human-producible adversarial examples, which are achieved by hand drawing. Specifically, in the digital space, the adversarial examples are optimized based on gradients and genetic search within a parameter space of line-based adversarial tags. Then, humans are asked to reproduce the adversarial lines given the (printed) digital image as the reference.

**Strengths:**

- The concept of “human-producible adversarial examples” was previously not well studied.
- The presentation of the paper is good, including sufficient example visualizations and clearly described technical details.
- Real-world experiments with human studies are conducted, involving both the tests with drawing and pasting tapes.

**Weaknesses:**

- Lack of motivation. The reviewer is not convinced about why “human-producible adversarial examples” are desired in the first place, given that electronic devices, e.g., printers and scanners, are quite easy to get nowadays. More importantly, printed adversarial examples have been shown to be very robust in the literature and the proposed  “human-producible adversarial examples”  have not saved any computational cost at all.
From the perspective of the budget of the attacker vs. defender, it seems not reasonable to constrain the attacker to use only pens since the defender has the ability to use a model/camera/scanner.
More specifically, in this paper, the authors indeed printed out those adversarial images with lines and let humans reproduce them. In this case, why don’t the authors directly use printed images as the final adversarial examples? By the way, these printed adversarial images should be compared as a baseline (non-human-producible) attack.

- Lack of discussion about potential countermeasures. A clear drawback of using a simple adversarial manipulation is the lack of stealthiness. In the case of line-based adversarial tags, it seems very easy to filter out them, e.g., based on edge detection or simply prototype matching. The authors should better test these potential countermeasures because attack stealthiness is important.

- As can be seen from the visualizations, the robust loss leads to lines quite close to the cup but the non-robust loss leads to lines further away. Why does it happen? If this really the case about why the robust loss works better, is it possible to just constrain the modification space to be surrounding the cup to make the attack robust?

- The detailed image design was not introduced until Section 3.5 “To test this real-world scenario, we conducted an experiment whereby we took photographs of a common household object – a cup – and produced an adversarial tag for them. We constrained the search area to a rectangular bounding box to limit the lines to a specific area of the image to avoid the cup itself.” However, without this introduction, it is very hard to understand the specific design depicted in Figure 2. The reviewer initially thought there was a sub-figure showing a cup pasted in the main figure showing the wall. So the suggestion is, for Figure 2, to detail those two factors independently but leave out the messages about the image design. By the way, why doesn’t the black area occupy the whole image?

- The use of “real world”/“physical world”. In the experiments, the authors consider drawing lines and pasting tapes. It is better to separate these two settings using more specific terms beyond the “real world”/ “physical world”.

**Questions:**

See the above weaknesses.

---

> ### Author Response · Authors · 2023-11-22
>
> We thank the reviewer for their comments. We will address the reviewer’s points in order:
>
> ```
> Lack of motivation. The reviewer is not convinced about why “human-producible adversarial examples” are desired in the first place, given that electronic devices, e.g., printers and scanners, are quite easy to get nowadays. More importantly, printed adversarial examples have been shown to be very robust in the literature and the proposed “human-producible adversarial examples” have not saved any computational cost at all. From the perspective of the budget of the attacker vs. defender, it seems not reasonable to constrain the attacker to use only pens since the defender has the ability to use a model/camera/scanner. More specifically, in this paper, the authors indeed printed out those adversarial images with lines and let humans reproduce them. In this case, why don’t the authors directly use printed images as the final adversarial examples? By the way, these printed adversarial images should be compared as a baseline attack.
> ```
>
> The authors tried but did not find any baseline appropriate to the threat model that the paper is addressing. Overall, due to our proposed attack only requiring a marking pen or a roll of tape, it is more portable and has a high access factor compared to methods that require printers or carrying around suspicious prefabricated patches. We can in principle report the numbers from other papers side by side but find this to be apples to oranges and not really representative of overall attack usefulness.
>
> Human-producible Adversarial Examples do not pursue cutting down computational costs, but rather focus on producing adversarial examples that humans can use without additional training or sophisticated equipment. It is not at all easy to manufacture adversarial glasses or an adversarial tshirt, but we find that it is trivial for humans to apply lines, making it a viable attack in a low-budget scenario where additional tools are not available.
> ```
> Lack of discussion about potential countermeasures. A clear drawback of using a simple adversarial manipulation is the lack of stealthiness. In the case of line-based adversarial tags, it seems very easy to filter out them, e.g., based on edge detection or simply prototype matching. The authors should better test these potential countermeasures because attack stealthiness is important.
> ```
>
> We thank the reviewer for pointing this out. We believe countermeasures such as edge detection or prototype-match would be very application-specific to be useful as lines occur often in the real world. In the same way as all of the other work, e.g. adversarial tshirts and glasses, we leave defences to future work.
>
> ```
> As can be seen from the visualizations, the robust loss leads to lines quite close to the cup but the non-robust loss leads to lines further away. Why does it happen? If this really the case about why the robust loss works better, is it possible to just constrain the modification space to be surrounding the cup to make the attack robust?
> ```
>
> We do not observe any common traits or patterns emerging from the robust and non-robust loss on similar images, and believe that the observations pertaining to lines closer to the subject leading to more robust loss are just coincidences. We believe that, much like for classic pixel-based adversarial example generation, there would be no human-discernable patterns that emerge.
>
> ```
> The detailed image design was not introduced until Section 3.5 “To test this real-world scenario, we conducted an experiment whereby we took photographs of a common household object – a cup – and produced an adversarial tag for them. We constrained the search area to a rectangular bounding box to limit the lines to a specific area of the image to avoid the cup itself.” However, without this introduction, it is very hard to understand the specific design depicted in Figure 2. The reviewer initially thought there was a sub-figure showing a cup pasted in the main figure showing the wall. So the suggestion is, for Figure 2, to detail those two factors independently but leave out the messages about the image design. By the way, why doesn’t the black area occupy the whole image?
> ```
> We acknowledge the confusing order of these two parts of the paper and now rephrase the sections. The explored black area is an artefact of the attack process line placement, as well as, jitter.
>
> ```
> The use of “real world”/“physical world”. In the experiments, the authors consider drawing lines and pasting tapes. It is better to separate these two settings using more specific terms beyond the “real world”/ “physical world”.
> ```
> We thank the reviewer for bringing this to our attention. We are not entirely sure what the reviewer sees as the difference between the settings as both drawing lines and pasting tape achieve the same results of introducing the black lines into the real world. Otherwise, we use the real and physical world interchangeably.

---

### Official Review · Reviewer_zAVL · 2023-10-31

**Soundness:** 2 fair
**Presentation:** 2 fair
**Contribution:** 2 fair
**Rating:** 3
**Confidence:** 3

**Summary:**

This paper proposes a new type of adversarial attacks, named adversarial tags. This attack can be accomplished in real world with only one marker pen. The authors propose generate-and-prune method to achieve this goal.

**Strengths:**

- The paper's central concept—developing adversarial examples that can be created with something as simple as a single marker pen—is intriguing. This raises significant security concerns, particularly the possibility of attackers drawing lines on the ground to mislead autonomous driving systems.
- The authors suggest employing Jitter and Erasing as augmentation techniques during the optimization process of adversarial examples. These methods could potentially enhance the robustness of adversarial examples when faced with real-world conditions.

**Weaknesses:**

- The most significant shortcoming of this paper is the extremely insufficient evaluations. Attacks like the ones proposed are designed to be executed in real-world scenarios, which are typically "black-box" in nature. To only assess such attacks in "white-box" settings is not adequate; after all, any attack designed to maximize the loss function might perform well against known classifiers in such a setting.
- Additionally, the experimentation is conducted exclusively on the YOLOv8 model. To ensure a thorough understanding of the attack's efficacy, it's crucial to extend the evaluations to include a broader range of models.
- In real-life conditions, adversarial implementations may face various distortions due to camera angles or environmental interferences like weather. However, the paper's evaluation seems too idealistic. The authors have chosen an optimal camera angle, and there is a noticeable absence of corruption in the adversarial tags. This approach doesn't fully simulate the real-world conditions where such adversarial techniques would be applied.

**Questions:**

To convincingly demonstrate the practical viability of the proposed attack, I recommend conducting more comprehensive evaluations. For instance, transitioning from white-box to black-box settings would better reflect real-world operational conditions. Additionally, incorporating transformations or Gaussian noise could simulate the kind of corruptions one might encounter in actual scenarios, thereby providing a more robust validation of the attack's effectiveness.

---

> ### Author Response · Authors · 2023-11-22
>
> We thank the reviewer for their comments. We will address the reviewer’s points in order:
>
> ```
> The most significant shortcoming of this paper is the extremely insufficient evaluations. Attacks like the ones proposed are designed to be executed in real-world scenarios, which are typically "black-box" in nature. To only assess such attacks in "white-box" settings is not adequate; after all, any attack designed to maximize the loss function might perform well against known classifiers in such a setting.
> Additionally, the experimentation is conducted exclusively on the YOLOv8 model. To ensure a thorough understanding of the attack's efficacy, it's crucial to extend the evaluations to include a broader range of models.
> ```
>
> We acknowledge the limited evaluation in the paper. We intended the paper to be a capability one. We now include black box transferability results in Appendix in Tables 2 and 3, but also would like to justify our choice of white-box evaluation on YoloV8. YoloV8 and the yolo family of models is one of the most widely-deployed ML models for vision and is often used out-of-the-box. Hence, we feel that the results of the white-box evaluation fit the realistic physical-world threat model outlined in the paper.
>
> ```
> In real-life conditions, adversarial implementations may face various distortions due to camera angles or environmental interferences like weather. However, the paper's evaluation seems too idealistic. The authors have chosen an optimal camera angle, and there is a noticeable absence of corruption in the adversarial tags. This approach doesn't fully simulate the real-world conditions where such adversarial techniques would be applied.
> ```
>
> We address the potential corruption and transformation misalignments that may come about due to improper recreation of the lines by introducing the jitter and erasure factors in the robust loss calculation. We refer the reviewer to Section 3.2 and Figure 2 that explains it in detail.
>
> As shown in the user study, these significantly improve the real-world efficacy of the attack. As we note in the paper, in practice other corruptions can be introduced alongside to gain invariance of choice e.g. lighting or angles.

---

### Official Review · Reviewer_HZPa · 2023-10-31

**Soundness:** 2 fair
**Presentation:** 4 excellent
**Contribution:** 2 fair
**Rating:** 3
**Confidence:** 4

**Summary:**

Adversarial examples have shown to be effective against DNNs, but require special tools or equipment to apply in the physical world. The authors investigate an intuitive subset of adversarial attack, which are those easily drawn by human hands. The methodology is a hybrid of gradient-free and gradient-based computation. The authors gradually tune the 4-tuple parameter set for each line in a collection given the gradient of the model with respect to an adversarial loss. The adversarial loss takes into account the range of intrinsic human error when drawing lines. A fixed amount of lines are pruned each iteration, iterating until the final set of adversarial lines is achieved. The authors conduct a series of experiments to check the effectiveness of the adversarial lines. A user study wih four participants was conducted to check the variance between non-robust and robust adversarial losses among users, as well as the variance among users on the same image. An experiment also measures the practical use-case of the lines, where everything except the object in focus is allowed to be modified by black tape. The experiments show that fewer long lines are more efficient for perturbation than many short lines.

**Strengths:**

* The writing quality is high and only a few minor typos are noticeable.
* The investigated problem is interesting and well-separated from existing literature. The plausible implementation of adversarial distortion by humans is still an under-studied area. The submission may offer some impact to the literature.
* The paper is well organized and flows easily from section to section.
* Experiments are intuitive and investigate key aspects of the methodology. The authors attempted a user study which gives some promising results.  The technique itself can reliably flip labels with up to 94% success.
* It was interesting to see the tradeoff between quantity and size of lines. Intuitively, more lines leads to more adversarial success, but in fact fewer long lines is more beneficial from a human factors standpoint. Likewise, more lines leads to more error when scanning the photos.
* The details of incorporating human error into the adversarial loss are useful to understand the potential attack surfaces of the model from a human-in-the-loop.
* The submission is headed in a good direction, but requires some more work before meeting the bar for publication (see Weaknesses).

**Weaknesses:**

* The user study consisted of only four participants, and does not investigate the effects of artistic ability. The sample size seems too low to draw any broad conclusions. It wasn't mentioned if the participants had a limited time budget to replicate the lines, which might be an important consideration in replication.
* The evaluation only considers a single YoloV8 classifier, rather than checking on multiple architectures and robustness levels, which are valid in the author's threat model. I was expecting experiments on robustified models [1,2] or some experimental results in trying to use adversarial lines for adversarial training. It would be interesting to see if the data generated by adversarial lines is too noisy, potentially degrading the benign accuracy when used for AT. This could provide useful information for the broader community.
* Only the white-box implementation is investigated, so the transferability of lines is unknown. It seems unlikely that a human with only a marker can also backprop through a service provider's model. I was expecting an experiment where the adversary tries to transfer adversarial lines from an owned model to an unseen model, potentially of a different architecture. The authors primarily pitch the contribution as an easily accessible technique for non-experts, so it seemed contradictory that they would also require access to weight-level knowledge of the defender's model. In the same spirit, the computational complexity seems too high for a non-expert to perform these attacks, since it requires a 4-GPU workstation to run.
* Only 500 images from 1000-class ImageNet were investigated (i.e., only half of classes are represented), and in that regard, the authors have only performed experiments on ImageNet. It isn't clear if the proposed methodology is applicable to other datasets, or how the attack behaves across different object classes. I suspect some object classes and camera angles are more difficult to attack under this threat model. This would change the overall feasability of the attack.
* The physical realization of attacks still seems unlikely, since most experiments allow adversarial lines to occupy any portion of the image, even the background. Previous work have successfully launched similar attacks by only perturbing the spatial region of a relevant object (e.g., clothing patch or fashion attacks).
* It is difficult to gauge the significance of the submission without comparisons to baseline techniques. For example, it seems feasible that existing white-box attacks could be used for a similar style attack by limiting their influence to the regions with adversarial lines, and limiting the fidelity of perturbation. Likewise, it isn't clear how adversarial lines perform compared to techniques such as adversarial clothing or sunglasses.


[1] Certified Adversarial Robustness via Randomized Smoothing. http://arxiv.org/abs/1902.02918

[2] Towards Deep Learning Models Resistant to Adversarial Attacks. http://arxiv.org/abs/1706.06083

**Questions:**

* It isn't clear why the adversary could only perturb the background of an image, rather than the object. This seems to go counter to previous work. Can the authors comment on the attack feasability from only perturbing the object's spatial region?
* Can the authors comment on the attack transferability? Is it feasible that adversarial lines would transfer to an unseen model?
* In the human study, how much time were participants given to replicate the lines?
* How does the proposed attack compare to existing physical attacks?
* What is the time complexity of running the attack on a single image? Does it scale well with the number of lines?
* How well do robust models fare against the adversarial lines? Can adversarial lines be used for adversarial training?
* Do the authors normalize the size of the lines for the size of the object? A black marker will cover more of a plastic cup, but not so much of a soccer ball.

---

> ### Author Response · Authors · 2023-11-22
>
> We thank the reviewer for their comments. We will address the reviewer’s points in order:
>
> ```
> The user study consisted of only four participants, and does not investigate the effects of artistic ability. The sample size seems too low to draw any broad conclusions. It wasn't mentioned if the participants had a limited time budget to replicate the lines, which might be an important consideration in replication.
> In the human study, how much time were participants given to replicate the lines?
> ```
>
> We have subsequently recruited 10 additional users with the same setting and observed consistent results. Throughout the user study, we did not enforce a time budget and asked the participants to replicate lines as flexibly as they liked. The timings are now presented in Appendix D of the paper, and show that even for 12 lines, users were able to create the adversarial tag in under 20 seconds. We did not investigate artistic ability because our focus is to evaluate the robustness of replication, and having artistic ability would only make the replication more precise which goes against our intention to evaluate the "bad" replication cases. Results are updated in the paper.
>
> ```
> The evaluation only considers a single YoloV8 classifier, rather than checking on multiple architectures and robustness levels, which are valid in the author's threat model. I was expecting experiments on robustified models [1,2] or some experimental results in trying to use adversarial lines for adversarial training. It would be interesting to see if the data generated by adversarial lines is too noisy, potentially degrading the benign accuracy when used for AT. This could provide useful information for the broader community.
> How well do robust models fare against the adversarial lines? Can adversarial lines be used for adversarial training?
> ```
>
> First, we want to note that robustness literature referred to is either rarely used in practice because of latency overheads e.g. with randomised smoothing, or incurs significant accuracy degradation e.g. with adversarial training. Second, even if we assume that adversarial training or smoothing is employed, the perturbation budget of the lines is significantly larger than anything that certification in current literature may provide. We now include Table 2 and 3 in Appendix B that show that we utilise perturbation with Linf norms from 214/255 to 242/255, compared to common certification of 8/255.
>
> ```
> Only the white-box implementation is investigated, so the transferability of lines is unknown. It seems unlikely that a human with only a marker can also backprop through a service provider's model. I was expecting an experiment where the adversary tries to transfer adversarial lines from an owned model to an unseen model, potentially of a different architecture. The authors primarily pitch the contribution as an easily accessible technique for non-experts, so it seemed contradictory that they would also require access to weight-level knowledge of the defender's model. In the same spirit, the computational complexity seems too high for a non-expert to perform these attacks, since it requires a 4-GPU workstation to run.
> Can the authors comment on the attack transferability? Is it feasible that adversarial lines would transfer to an unseen model?
> ```
>
> We acknowledge the limited evaluation in the paper. We intended the paper to be a capability one. We now added the black box transferability results to Table 4 in Appendix C – we do find evidence that adversarial examples transfer across the yolo family.
>
> We next would like to justify our choice of white-box evaluation on YoloV8. YoloV8 and the yolo family of models is one of the most widely-deployed ML models for vision and is often used out-of-the-box. Hence, we feel that the results of the white-box evaluation fit the realistic physical-world threat model outlined in the paper.
>
> We clarify that “capability” refers to the user’s capability of realizing a given adversarial perturbation. Hence, the computational cost for computing the perturbation is outside the scope of a user’s capability, e.g., by deploying the attack on an easy-to-access cloud server. We consider the retrieval and realization of adversarial perturbation to be two disentangled stages. At the same time, the method itself is not particularly memory or GPU-heavy and could likely be run on a modern smartphone, albeit with a larger time budget.

---

> > ### Author Response · Authors · 2023-11-22
> >
> > ```
> > Only 500 images from 1000-class ImageNet were investigated (i.e., only half of classes are represented), and in that regard, the authors have only performed experiments on ImageNet. It isn't clear if the proposed methodology is applicable to other datasets, or how the attack behaves across different object classes. I suspect some object classes and camera angles are more difficult to attack under this threat model. This would change the overall feasability of the attack.
> > ```
> > We acknowledge the inappropriately-small size of the evaluation set. This is due to limited computing resources. We are now in the process of expanding our evaluation set to be a larger, more complete subset of ImageNet.
> >
> > ```
> > The physical realization of attacks still seems unlikely, since most experiments allow adversarial lines to occupy any portion of the image, even the background. Previous work have successfully launched similar attacks by only perturbing the spatial region of a relevant object (e.g., clothing patch or fashion attacks).
> > It isn't clear why the adversary could only perturb the background of an image, rather than the object. This seems to go counter to previous work. Can the authors comment on the attack feasability from only perturbing the object's spatial region?
> > ```
> > We clarify that “perturbing the spatial region of a relevant object” is an easier attack setting, as such attacks can directly change the appearance of the object of interest. In contrast, our motivation is to reduce the capability requirement on the human side and avoid perturbing the object itself.
> > ```
> > It is difficult to gauge the significance of the submission without comparisons to baseline techniques. For example, it seems feasible that existing white-box attacks could be used for a similar style attack by limiting their influence to the regions with adversarial lines, and limiting the fidelity of perturbation. Likewise, it isn't clear how adversarial lines perform compared to techniques such as adversarial clothing or sunglasses.
> > How does the proposed attack compare to existing physical attacks?
> > ```
> > The authors did not find any baseline appropriate to the threat model that the paper is addressing. Overall, due to our proposed attack only requiring a marking pen or a roll of tape, it is more portable and has a high access factor compared to methods that require printers or carrying around prefabricated patches. We can in principle report the numbers side by side but find this to be apples to oranges and not really representative of overall attack usefulness.
> >
> > ```
> > Do the authors normalize the size of the lines for the size of the object? A black marker will cover more of a plastic cup, but not so much of a soccer ball.
> > ```
> > The size of the lines are pre-determined at attack time and are guestimated by eye to match whatever surface and marker/tape that we are planning on using in the physical investigation.
> > ```
> > What is the time complexity of running the attack on a single image? Does it scale well with the number of lines?
> > ```
> > The average number of steps to flip the top-1 prediction is presented in Figures 4 and 5 in the paper. The exact timings depend very heavily on the compute available, however we have run an additional experiment with two timing groups - fewer lines (between 3 and 6 inclusive) and more lines (between 8 and 11 inclusive). Both groups used robust loss, which is significantly more compute intense. This test was performed on a single RTX2080Ti. The results were as follows. The group with [3, 6] lines took an average of 2050.5s (34.2min) to run with a standard deviation of 1663.2s (27.7min). The group with [8, 11] lines took an average of 2561.6s (42.7min) with a standard deviation of 2987.3s (49.8min). The attack was considered successful (and hence stopped) if the top-1 prediction of the image changed under the influence of the adversarial tags.

---

### Official Review · Reviewer_e2cA · 2023-11-01

**Soundness:** 3 good
**Presentation:** 2 fair
**Contribution:** 2 fair
**Rating:** 5
**Confidence:** 4

**Summary:**

This paper presents a compelling approach, suggesting the use of line-based adversarial tags to deceive the predictions of YOLO-based models. Notably, the authors have taken an intriguing step by ensuring that these adversarial lines can be realistically produced by humans, adding a layer of practicality to their proposal.

**Strengths:**

1. This paper delves into a captivating avenue for generating adversarial perturbations.
2. Moreover, the authors have thoughtfully crafted a robust loss function aimed at ensuring that the adversarial lines are feasibly replicable by humans.

**Weaknesses:**

1. The "Method" section would benefit from additional granularity. Specifically, it remains unclear how overlapping of the randomly generated lines is addressed. Are there any constraints or specific guidelines governing the generation of these random lines?
2. The decision to employ only four line-defining points warrants clarification. Is there a theoretical foundation or empirical rationale that supports this choice?
3. There are noticeable writing inconsistencies. For instance, the abbreviation "NLL" is employed prior to its formal definition, which might be confusing for readers.

**Questions:**

This paper could be significantly enhanced by addressing the following queries:

1. What underlying principles or mechanisms allow simple lines to effectively execute adversarial attacks?
2. When considering both targeted and untargeted attacks on images of similar objects, what common traits or patterns emerge?
3. The motivation presented could be better articulated. What is the rationale behind necessitating human production of the adversarial line? Is there an inherent advantage or specific scenario where this becomes crucial?
4. In scenarios where the lines are confined solely within the object's area, does the efficacy of the proposed attack remain consistent?

---

> ### Author Response · Authors · 2023-11-22
>
> We thank the reviewer for their comments. We will address the reviewer’s points in order:
>
> ```
> The "Method" section would benefit from additional granularity. Specifically, it remains unclear how overlapping of the randomly generated lines is addressed. Are there any constraints or specific guidelines governing the generation of these random lines?
> ```
>
> We did not enforce particular constraints such as non-overlapping on the generated lines, as overlapping lines are still replicable in the real world. The generation of lines includes two stages. First, we sample adversarial lines uniformly at random over the full extent of the image (or a rectangular subset if specified). Second, we iteratively adjust the position of these lines via gradient descent. The gradient is calculated directly on the rasterized lines following "Differentiable Drawing and Sketching" (Daniela Mihai and Jonathon Hare, 2021).
>
> ```
> The decision to employ only four line-defining points warrants clarification. Is there a theoretical foundation or empirical rationale that supports this choice?
> ```
> We outline in section 4.1 that the method can be applied to Bezier curves and more complicated shapes, but we restrict our investigation to straight lines, which can be uniquely defined by their start and end coordinates, which is where the 4 parameters come from.
>
> ```
> There are noticeable writing inconsistencies. For instance, the abbreviation "NLL" is employed prior to its formal definition, which might be confusing for readers.
> ```
> We apologise for the inconsistencies and many thanks for pointing this out! It is now corrected in the manuscript.
>
> ```
> What underlying principles or mechanisms allow simple lines to effectively execute adversarial attacks?
> When considering both targeted and untargeted attacks on images of similar objects, what common traits or patterns emerge?
> ```
> Adversarial examples still puzzle researchers 10 years after their discovery; we also, unfortunately, still do not have a solution to them. In our work we highlight that something as simple as a few lines can be enough to disrupt such a sophisticated model as YOLO; importantly, with our work humans can trick YOLO effectively.
>
> We follow the attack process that is common in the adversarial example literature. We explicitly search for adversarial perturbation that maximises the classification loss within a space represented by lines. The attack’s success suggests that the classic pixel-based adversarial perturbation also exists in a far simpler space represented by lines. Our findings are also consistent with existing adversarial examples literature in that we do not observe any common traits or patterns emerging from targeted and untargeted attacks on similar images.
>
> ```
> The motivation presented could be better articulated. What is the rationale behind necessitating human production of the adversarial line? Is there an inherent advantage or specific scenario where this becomes crucial?
> ```
>
> We acknowledge the reviewer’s concerns regarding the motivation of the paper.
>
> Attacks are categorised by their costs. Usually more sophisticated attacks come with higher costs, making them harder to use in practice. We see our work as a natural continuation of physical adversarial examples literature, where we now enable faster (e.g. require application of a few lines, humans manage it in seconds), deniable (e.g. its just a few lines outside of the object, no odd looking adversarial patches, an attacker doesn't need to carry around suspicious looking patches), cheaper (e.g. drawing pens are available everywhere, graffiti pens are used all across the world), and usable attacks (e.g. the first attack of its kinda that is easily applicable by human, as is proven by our human evaluation).
>
> ```
> In scenarios where the lines are confined solely within the object's area, does the efficacy of the proposed attack remain consistent?
> ```
>
> We find that the attack remains successful when confining the lines within the object’s area. Additionally, we clarify that constraining to the object area is an “easier” attack, as it can easily change the existing object’s appearance to fool the classifier. Our setting is arguably more challenging because we need to change the model’s output without perturbing the object itself, but the background around it.

---

### Meta-Review · Area_Chair_R5hR · 2023-12-11

**Metareview:**

The paper presents a method for generative human-producible adversarial examples (requiring only a marker pen). The paper received four reviews. All reviews are leaning negative. The main drawbacks outlined by the reviewers are
- limited user study (only 4 participants)
- only YOLOv8 classifier, no test on the adversarial transferrability
- only white box is tested and only 500 images on imageNet tested.
- no simulation of the real-world conditions.

Overall, the practical viability of the proposed attack is not convincingly validated. The authors did rebut some of the raised concerns. Unfortunately, the AC does not feel that they adequetely address the issues of insufficient evaluation. Given the concensus among the reviewers, the AC finds no ground to accept.

**Justification For Why Not Higher Score:**

All the reviewers vote to reject.

**Justification For Why Not Lower Score:**

N/A

---

### Decision · Program_Chairs · 2024-01-16

Reject